Tomato yellow leaf curl virus manipulates Bemisia tabaci, MEAM1 both directly and indirectly through changes in visual and volatile cues

Paris Thomson M. 1
Johnston Nicholas 1
Strzyzewski Iris 1
Griesheimer Jessica L. 1
Reimer Benjamin 1
Malfa Kathi 1
Allan Sandra A. 2
Martini Xavier xmartini@ufl.edu 1
1 Department of Entomology and Nematology, North Florida Research and Education Center, University of Florida , Quincy , FL , United States of America
2 Insect Behavior and Biocontrol Research Unit, USDA-ARS , Gainesville , FL , United States of America
Gawande Suresh
Electronic publication date: 2024 Jul 23
Publication date: 2024
Volume: 12
Electronic Location ID: e17665
Received 2024 Mar 15; Accepted 2024 Jun 11
Copyright: ©2024 Paris et al.
Copyright year: 2024
Copyright holder: Paris et al.
License: This is an open access article distributed under the terms of the Creative Commons Attribution License, which permits unrestricted use, distribution, reproduction and adaptation in any medium and for any purpose provided that it is properly attributed. For attribution, the original author(s), title, publication source (PeerJ) and either DOI or URL of the article must be cited.
License URL: https://creativecommons.org/licenses/by/4.0/

Keywords: Whitefly, Methyl salicylate, Volatile organic compounds, Host manipulation

Funding: Florida Department of Agriculture and Consumer Service #027426 The USDA National Institute of Food and Agriculture, Hatch project FLA-NFC-006275 Southern SARE through a graduate student GS19-210 The National Science Foundation (NSF), IUCRC program (CAMtech grant) All the external funding received during this study consisted of: (1) A grant awarded by the U.S. Department of Agriculture’s (USDA) Agricultural Marketing Service through the Florida Department of Agriculture and Consumer Service (Specialty Crop Block Grant #027426), (2) by the USDA National Institute of Food and Agriculture, Hatch project FLA-NFC-006275, (3) by Southern SARE through a graduate student grant (GS19-210), (4) by the National Science Foundation (NSF), IUCRC program (CAMtech grant). The funders had no role in study design, data collection and analysis, decision to publish, or preparation of the manuscript.

==============================
The sweetpotato whitefly, Bemisia tabaci MEAM1, is one of the most devastating pests of row-crop vegetables worldwide, damaging crops directly through feeding and indirectly through the transmission of many different viruses, including the geminivirus Tomato yellow leaf curl virus (TYLCV). Y-tube olfactometer tests were conducted at different stages of TYLCV infection in tomatoes to understand how TYLCV affects B. tabaci behavior. We also recorded changes in tomato hosts’ color and volatile profiles using color spectrophotometry and gas chromatography-mass spectrometry (GC-MS). We found that the infection status of B. tabaci and the infection stage of TYLCV influenced host selection, with uninfected whiteflies showing a preference for TYLCV-infected hosts, especially during the late stages of infection. Viruliferous B. tabaci attraction to visual targets significantly differed from non-viruliferous B. tabaci. Late-stage infected hosts had larger surface areas reflecting yellow-green wavelengths and higher emissions of methyl salicylate in their volatile profiles. These findings shed new light on several critical mechanisms involved in the viral manipulation of an insect vector and its economically important host.

Introduction

The silverleaf whitefly, Bemisia tabaci MEAM1, is a polyphagous agricultural pest found worldwide, causing considerable economic damage to many commercially valuable crops via feeding damage and transmitting many different viruses (De Barro et al., 2011). One such begomovirus, Tomato yellow leaf curl virus (TYLCV), is transmitted primarily by B. tabaci and is considered the most harmful insect-transmitted disease of tomatoes worldwide. Since most whiteflies have evolved close mutualistic relationships with the plant pathogens they harbor, understanding how these interactions influence the insect vector’s behavior is crucial for addressing current problems the agricultural industry is facing regarding pest management and mitigation of viruses such as TYLCV.

Plant pathogen manipulation of insect vector behavior is a phenomenon that has come under increasing scrutiny in recent years. Plant pathogens may indirectly influence insect behavior through changes in plant phenotype or directly influence insect behavior once the insect acquires the pathogen by feeding on infected plant tissues (Mauck, De Moraes & Mescher, 2016; Eigenbrode, Bosque-Pérez & Davis, 2018). Plant pathogens’ indirect manipulation of insect vectors can occur by changing the visual or volatile profiles of the insect’s host plant. For example, Candidatus L. asiaticus, the phytopathogen that causes citrus greening disease, increases the production of methyl salicylate (MeSA) in infected citrus, which recruits higher numbers of Asian citrus psyllids to feed on the infected tissue and acquire the pathogen (Mann et al., 2012; Martini et al., 2016). One can also see the importance of visual cues in pathogen manipulation in examples such as how the Tobacco etch virus induces chlorotic mottling of plant tissue, which visually increases the tobacco plant’s attractiveness to aphids (Eckel & Lampert, 1996). Understanding the nuances of the tri-trophic interactions between insect vectors, plant pathogens, and plant hosts requires a careful examination of how each component influences the other to understand the disease pathosystem fully.

Like many phytophagous insects, host-searching behavior in B. tabaci is driven by a combination of visual and olfactory cues. To propagate, viruses such as TYLCV can take advantage of these sensory-mediated interactions to manipulate uninfected whiteflies into feeding on virus-infected tissue (Fang et al., 2013). For example, yellow-green wavelengths of light visually attract B. tabaci and the symptoms of infected plants (chlorotic and yellowing tissue) make infected tomato plants more visually appealing to whiteflies (Johnston & Martini, 2020). In addition to visual cues, TYLCV changes the volatile profile emitted from an infected tomato plant, which may also influence B. tabaci’s host selection behavior. However, the extent of this behavioral shift remains unclear (Chen et al., 2017). Finally, studies have shown that the acquisition of the virus by its whitefly vector directly influences B. tabaci behavior, with infected whiteflies tending to shift their host feeding preferences towards healthy plants after acquiring the virus (Moreno-Delafuente et al., 2013; Liu et al., 2013; Fang et al., 2013). This mechanism of inducing behavioral change in its vector is yet another way TYLCV ensures its propagation. Despite a basic understanding of the TYLCV disease cycle, there is little documentation on how TYLCV alters tomato host plants in the visual spectrum and through discrete semiochemicals to maximize their appeal to healthy B. tabaci.

In this study, we evaluate how TYLCV impacts the host choice behavior of B. tabaci, considering both the stage of plant infection and the infection of the insect vector itself. To further understand how TYLCV visually manipulates whitefly phototaxis, host choice and topical analysis of infected leaf tissue were also conducted at different stages of infection using color spectrometry. Additionally, by presenting viruliferous and uninfected B. tabaci with a two-choice yellow and green visual target, we investigated the effect of TYLCV on the behavior of B. tabaci towards visual cues. We also examined changes in the profiles of tomato volatiles using gas chromatography-mass spectrometry (GC-MS) and correlated these changes with shifts in whitefly host choice behavior over time. Understanding these critical interactions between host, vector, and plant pathogens will help unlock new methods for combatting TYLCV and contribute to applying more timely pest management strategies.

Materials and Methods

Insect colonies

Wildtype specimens of Bemisia tabaci MEAM1 were collected initially from infested straightneck squash, Cucurbita pepo L. cv. recticollis, in Quincy, Florida, in fall 2018. Whitefly biotype testing was conducted by genetic analysis following the published protocol by McKenzie & Osborne (2017). We then divided the collected whiteflies into two separate colonies: one with TYLCV-infected tomato plants and insects and one without TYLCV. We reared whiteflies on collard, Brassica oleracea, of the Flash variety (Seedway® 674580NT) for the uninfected colony in a small greenhouse. We placed several sentinel tomato plants in the greenhouse and periodically tested them for the presence of the virus using traditional PCR to ensure that TYLCV was absent in the uninfected colonies. The whiteflies used for the 2-choice assay involving visual targets were reared in laboratory conditions (22.1 ± 0.23 °C, 42.9 ± 1.3% RH) in screen cages on 5–10 collard plants under a 16 h L:D photoperiod.

Hybrid tomatoes, Solanum lycopersicum, of Florida 47 R variety (Seedway® 818212NT), were utilized in both whitefly colonies. Methods for insect maintenance were similar to those described in Johnston & Martini (2020). Tomato seeds were planted on plastic trays (51 × 25 cm) containing agricultural substrate (Jolly Gardener Proline C/B Growing Mix). Seedlings were then transplanted into 11.5 cm2 black nursery pots (one seedling per pot) three weeks after sowing and maintained under greenhouse conditions. After one month, plants to be used for the experiments were moved to white mesh screen cages (40 × 40 × 60 cm) and inoculated them with TYLCV, by introducing to each cage 100 adult B. tabaci from the TYLCV whitefly rearing. After 14 days, PCR tests ran on a subset of plants confirmed that tomato plants were positive for TYLCV, and we isolated thirty of them in a small rearing chamber maintained at a constant temperature of 22 °C (12 h L: D period). After one month, a new batch of uninfected tomato plants was added to the infected batch and labeled them with the initial infection date. This process monthly was repeated to obtain a range of diseased plants, with the older plants showing the most severe TYLCV symptoms. It was assumed that all whiteflies collected on TYLCV-infected plants were also TYLCV-positive, having acquired the virus through feeding on infective tissue since 100% viral acquisition from tomatoes after 12 h of feeding has been demonstrated (Jiu, Zhou & Liu, 2006).

Olfactometer choice assays

Olfactometer assays were conducted with the same methods than in Johnston & Martini (2020). Whitefly choice tests between healthy tomato plant cues and TYLCV-infected tomato plant cues were conducted on three levels: (1) visual cues only, (2) olfactory cues only, and (3) visual + olfactory cues. Forty individual whiteflies were evaluated at each level for both TYLCV-infected and uninfected populations, resulting in a total of 80 recorded choices per level. In addition, whitefly preference to TYLCV-infected tomato plants was tested at three different stages of infection: (1) presymptomatic (pre), or plants inoculated with TYLCV two weeks prior that have yet to show symptoms (Fig. 1A), (2) mid-stage symptomatic (mid), or plants that show visible TYLCV symptoms approximately one month after inoculation (Fig. 1B), and (3) late-stage symptomatic (late), or plants that show severe TYLCV symptoms approximately two or more months after inoculation (Fig. 1C). For the experiments that only used visual cues, a glass Y-tube horizontally was positioned under a fluorescent light source mounted within a white fiberboard box for uniform light diffusion. Then, visual targets were placed (tomato leaves representing different levels of TYLCV disease) underneath each arm of the glass Y-tube and a single whitefly was released at the bottom of the tube (Johnston & Martini, 2020). Each whitefly had 15 min to choose between the two arms, where crawling at least one cm into either arm constituted a choice. The visual targets were switched to opposite arms after every two whitefly choices to ensure no bias between the right and left arm of the olfactometer.

Figure 1 Tomato leaves with one of three different stages of TYLCV.

Presymptomatic (A), mid stage symptomatic (B), and late stage symptomatic (C). The specific locations of the spectral reflectance measurements are indicated by the colored circles on each leaf. The colors of the circles and the letters beside the colored circles indicate color category into which the spectral measurements were placed. Yellow circles with the code Y represent yellow regions of the leaf. Green circles with the code G represent green regions of the leaf. Orange circles with the code DY represent darker yellow to brown areas of the leaf.

For experiments that only used odor cues, the odor source consisted of a single tomato plant, either TYLCV-infected or uninfected. Tomato plants were positioned within a cylindrical glass jar (16 × 50 cm) sealed with a Teflon lid (Sigma Scientific, Gainesville, FL, USA). Two outlets were placed on the Teflon lid with Teflon tubing connecting one outlet to the airflow source and the other outlet to a single arm of the Y-tube glassware. Glass jars received charcoal-purified and humidified air at 0.1 L/min from a custom-made air delivery system (Sigma Scientific, Gainesville, FL, USA). After allowing air to circulate for at least 2 min through the apparatus, a single whitefly was introduced to the bottom of the Y-tube and allowed to choose the same manner as the visual cues-only experiments. The third set of experiments that evaluated host choice under visual and olfactory cues combined the visual tomato leaf targets underneath the Y-tube glassware with the setup of the olfactory cues only experiments.

Since we reared uninfected whiteflies and TYLCV-infected whiteflies on different hosts (collard and tomato, respectively), we conducted additional control replicates to ensure that host rearing did not bias the final host selection. These control replicates examined the whitefly choice between collard and uninfected tomato volatiles using the Y-tube olfactometer setup previously described. In the control trials, 40 individual choices were made by whiteflies reared on collards and compared to 40 individual choices by whiteflies reared on uninfected tomato plants. Once we completed the control trials and confirmed the null hypothesis (see results), we used collard as the host-rearing plant for all non-viruliferous whiteflies for logistical reasons.

Y-Tube Methyl salicylate choice assays

Methyl salicylate (MeSA) volatiles increased significantly in TYLCV infected tomatoes (see results); therefore we conducted an experiment to test the response of whiteflies to different MeSA concentrations. MeSA (≥ 99% (GC), #M6752; Sigma-Aldrich) concentrations tested were created from a stock solution and subsequently serial diluted with a mineral oil base. We conducted whitefly choice tests between MeSA volatile cues at three concentrations: (1) 1.0 µg/µl MeSA vs. mineral oil, (2) 0.1 µg/µl MeSA vs. mineral oil, and (3) 0.01 µg/µl MeSA vs. mineral oil. We evaluated forty individual whitefly choices at each level using TYLCV-infected and uninfected populations, recording 80 whitefly choices at each level.The setup consisted of a glass Y-tube positioned vertically with a burette stand and clamp under a fluorescent light source mounted within a white fiberboard box for uniform light diffusion. A single cotton roll (Dynarex Corporation, #3250) was infused with 100 µL of one of the MeSA concentrations or the mineral oil control and added to an oven bag (Reynolds Kitchens® Oven Bags, 482 mm × 596 mm) baked at 60 °C overnight. We positioned two outlets on the oven bag, connecting one outlet to the airflow source and the other to a single arm of the Y-tube glassware with Teflon tubing. Teflon bags received charcoal-purified and humidified air at 0.1 L/min from a custom-made air delivery system. After allowing air to circulate for at least 2 min through the apparatus, a single whitefly was introduced to the bottom of the Y-tube and allowed to choose in the same manner as the visual cues only experiments.

Volatile collection

Data for the profile of TYLCV-infected tomato plant volatile odors were collected as previously described in Strzyzewski, Funderburk & Martini (2023). Specifically, the volatile collection system consisted of the top half of the tomato main stem enclosed within an oven bag (Reynolds, Lake Forest, IL, USA) and tied at the top and bottom with zip ties. The oven bag was baked overnight at 60 °C to reduce contaminants. Incoming air was purified via a charcoal filter and pushed in at the top of the oven bag at a rate of 1.0 L/min. The volatiles were forced to the bottom of the bag by pulling air at 0.5 L/min with a controlled vacuum from the automated volatile collection system through a volatile collection trap (7.5 cm long) with 30 mg of HayeSep Q adsorbent (Volatile Assay Systems, Rensselaer, NY, USA) connected to the bottom of the bag with a PTFE fitting. Volatiles were collected for 24 h, andvolatile samples were extracted in vials with 150 µl of dichloromethane. Beforehand, 1 µg of nonyl acetate as an internal standard was added to the samples. 1 µL of each sample was injected into GC-MS (Thermo Scientific ISQ) using an autosampler. Helium was used for the carrier gas at a two mL/min linear flow velocity. All samples were analyzed on a fused silica TG-5MS column (5% phenyl methyl polysiloxane) (Thermo Fisher Scientific, Waltham, MA, USA) 30 min × 0.25 mm ID. The column oven temperature was maintained at 40 °C for 1 min and increased at a rate of 7 °C/min to a final temperature of 300 °C and maintained at 300 °C for 6 min. The injector temperature was set at 270 °C, with the detector set at 200 °C. Quantitation was assigned by comparing peak areas of known amounts of internal standard, nonyl acetate, with the area under the peak of compounds extracted from the infected plants. Compounds were first tentatively identified by comparison of mass spectra with available mass spectra libraries (NIST 11, Wiley 9, mainlib), then confirmed for those available at a reasonable price by injection of analytical standards. Every compound in Table 1 was confirmed by GC-MS analyses with their corresponding synthetic compounds purchased from Sigma-Aldrich. There were three to five replicates per TYLCV-infection stage.

Two choice visual assay

The study assessed viruliferous and nonviruliferous whiteflies’ visual preferences when exposed to filtered light visual targets. Based on the results from the previous experiments, we tested whiteflies’ visual preference against yellow and green visual cues using a two-choice visual target test. The setup included an enclosed space with a base chamber and adhesive translucent panels at the top for capturing the whiteflies, after Paris et al. (2017). This enclosed space was a black cylindrical tube, 12 cm in diameter and 30 cm high, with a small chamber (4 cm by 6.5 cm) attached to its lower center. We coated the inside of the tube and chamber in matte black paint. We attached a shutter mechanism beneath the tube and inserted the chamber into this shutter. The chamber’s base was covered with a black plastic lid, measuring 4.5 cm in diameter and 3 cm in height.

Table 1 Compounds detected in the volatile emissions in a healthy tomato plant and infected tomatoes at three stages of TYLCV infection: pre-symptomatic, mid-stage, and latestage infection.

We calculated the internal standard (IS) based on nonyl acetate (RT = 19.36) injected under the same conditions as the samples. Pres. indicates the number of times the compound was detected in replicates. The ratio is the average ratio between the peak compound and the IS. Different letters indicate significant differences among the treatment for a specific compound. An asterisk (*) indicates compounds that were verified with analytical standards.

		Control tomatoes	Pre-symptomatic TYLCV	Mid-symptomatic TYLCV	Late-symptomatic TYLCV	Kruskal-Wallis	
RT	Compound	Pres.	Ratio	Pres.	Ratio	Pres.	Ratio	Pres.	Ratio		
8.48	α-pinene*	2/5	0.05 ± 0.03	1/3	0.12 ± 0.12	2/4	0.15 ± 0.11	2/5	0.04 ± 0.03	H = 0.52
P = 0.914	
9.57	3,7,7-trimethyl-1,3, 5-cycloheptatriene	2/5	0.16 ± 0.11
B	3/3	1.98 ± 0.40
A	4/4	1.78 ± 0.61
AB	4/5	1.26 ± 0.76
AB	H = 8.135
P = 0.043	
10.45	4-carene	4/5	3.09 ± 1.05	2/3	4.90 ± 2.47	4/4	8.53 ± 2.25	4/5	5.24 ± 3.02	H = 3.78
P = 0.286	
10.54	α-terpinene	4/5	1.44 ± 0.39	3/3	0.53 ± 0.36	3/4	0.50 ± 0.28	4/5	1.09 ± 0.53	H = 2.12
P = 0.549	
11.31	β-phellandrene	5/5	15.67 ± 3.32	3/3	12.19 ± 5.60	4/4	19.82 ± 2.01	5/5	14.72 ± 4.80	H = 1.43
P = 0.317	
16.17	Methyl salicylate*	0/5	0.00
B	1/3	0.12 ± 0.12
BA	3/4	0.11 ± 0.06
BA	5/5	2.19 ± 1.56
A	H = 9.25
P = 0.026	
22.37	Caryophyllene*	5/5	1.19 ± 0.12	3/3	1.16 ± 0.63	4/4	1.78 ± 0.37	5/5	3.42 ± 1.73	H = 2.23
P = 0.526	
23.20	Humulene	3/5	0.10 ± 0.06	3/3	0.72 ± 0.12	3/4	0.39 ± 0.15	2/5	0.60 ± 0.48	H = 6.063
P = 0.109	

A 400-watt metal halide lamp, positioned 55 cm above the enclosure, provided illumination. This lamp delivered around 2000 lux and emitted both UV and visible light. We covered the chamber’s top with an inverted, transparent Petri dish (150 mm in diameter and 15 mm high) and lined its interior base with a thin adhesive coating (Tangletrap1, Grand Rapid, MI, USA). Visual stimuli consisted of dyed polyester filters attached to the top of the inverted Petri dish, each covering half of the surface (either right or left side). To prevent ambiguity in determining the whiteflies’ color choice at the meeting point of the two colored filters in the middle of the Petri dish, we placed a strip of black electric tape (1.905 cm wide) across the dish’s diameter. We counted whiteflies that landed on the black electric tape portion as non-responders. We randomly alternated the placement of these filters (left or right side of the Petri dish) between experiments.

Plants infected with TYLCV showed altered reflectance spectra, primarily in the visible spectrum. Therefore, we chose the selected filters for their capacity to block UV while transmitting visible light. These included Roscolux1 yellow (#4530), Moss green (#89), UV block (#3114) with 94% transmission, and a neutral density filter (#3415) with 0.15 optical density (Rosco, Stamford, CT), cut to fit half of the Petri dish. To further impede UV light transmission through the colored filters, we layered UV-blocking filters atop the colored ones. Although these UV-blocking filters were primarily transparent to visible light, they significantly reduced UV transmission to less than 10% of its initial intensity. Figure 2 depicts the percent reflectance of each filter.

Figure 2 Transmission of filtered visual targets in the electromagnetic spectrum between 350 to 750 nm.

The yellow line represents visual target transmission of a yellow filter. The green line represents the visual target transmission of a green filter. Both lines represent the average of two spectrometer readings of the transmission of light between 350–750 nm.

For the experiments, we collected whiteflies into plastic vials (3 cm wide ×6 cm tall) with lids and fitted gaskets around the vials to ensure a snug fit in the assay arena’s holding chamber. Before starting an assay, we closed the manual shutter, tapped the vial to move the whiteflies to the bottom, removed the cap, and then quickly placed the vial into the holding chamber. A black coverslip was positioned on the chamber’s bottom, keeping the whiteflies in total darkness.

We dark-adapted the whiteflies for 10 min inside the holding chamber before each test. Then, we placed the filters on top of the test arena, allowed the whiteflies to adapt to the dark, opened the shutter, and allowed the whiteflies an hour to respond to the visual target. After the test, we considered the whiteflies stuck to the sticky layer as responders and determined their sex. We designated whiteflies that could not be identified to sex as unknown sex.

Whiteflies in the chamber or landing on the black line of the Petri dish’s sticky layer were labeled non-responders. We used CO2 to immobilize these non-responders for sex determination. We carried out the experiments between 08:00 and 18:00 h, with each test involving 20 to 30 whiteflies. We calculated the total response of whiteflies to either visual target by counting the whiteflies that went to either side of the visual target. In contrast, we counted non-responders as those that did not land on either visual target and were anywhere else in the arena. To ensure statistically representative proportions, we only included tests in the analysis where 40 percent or more of the whiteflies responded to the color choice of visual targets. We calculated the response as a proportion of the total number of responders. We repeated this process 25 times to replicate the assays.

Color spectrometry

We used a spectrometer (UV–VIS Black-Comet; StellarNet Inc, Tampa, FL, USA) to measure the spectral reflectance of tomato leaves at varying stages of TYLCV infection. The spectrometer, equipped with a focus lens, had seven exterior fibers illuminated by the light source and an interior fiber that enabled us to pinpoint the area of spectral reflectance. A Tungsten Halogen (350–2300 nm) and Deuterium (200–400 nm) light source were merged using a fiberoptic cable and equilibrated with filters. We used a small insect pin to create a hole in the leaf to record the locations where we made visual spectral reflectance readings (Fig. 1). The leaves were then photographed on a black background by a Nikon HDSLR (D5300) using an AF-P DX NIKKOR 18–55 mm f/3.5−5.6G VR lens. In Adobe Photoshop, we removed the background surrounding the leaves and replaced the holes made by the insect pin with colored digital circles. If the leaves were healthy control leaves, we took only three measurements at similar colored regions of the leaf. If the leaves were moderately expressing phenotypic characteristics of TYLCV, we took a single measurement at one of the three color regions if green, yellow, or dark yellow were present. The spectrometer produced electromagnetic spectrum values every 0.5 nm (Fig. 1). Researchers categorized the leaves by infection status, including presymptomatic (green throughout), mid-stage symptomatic (green, yellow, and dark yellow spots), and late-stage symptomatic (green, yellow, and dark yellow spots).

Statistical analysis

We conducted all statistical analyses using the statistical software R (RStudio Team, 2020; R Core Team, 2022). We used the following packages to run the analyses: readxl, readr, dplyr, lsr, car, citation, nlme, reshape2, multcomp, and emmeans (Pinheiro & Bates, 2000; Wickham, 2007; Hothorn, Bretz & Westfall, 2008; Navarro, 2015; Fox & Weisberg, 2019; Lenth, 2022; Wickham & Bryan, 2022; Wickham et al., 2022a; Wickham, Hester & Bryan, 2022b; Dietrich & Leoncio, 2023). We conducted a general linearized model (GLM) with binomial distribution for all replicates with plant stage of infection (early, mid, late) and whitefly viral status (positive, negative) treated as categorical factors to determine differences in whitefly choice. We also conducted a chi-square test as a post-hoc analysis for olfactometer choice experiments (α = 0.05).

For the spectrometer measurements, we divided the color region of the electromagnetic spectrum into two categories of interest: between 400 and 750 nm, 496 and 570.5 nm for green, and 571 and 590.5 nm for yellow. Using the Akaike information criterion (AIC), we modeled the percent reflectance of the leaves using the following variables: percent reflectance (p), wavelength in nanometers (λ), regression coefficient (a), leaf with level of exposure to TYLCV (lf), and color region of the electromagnetic spectrum (cr). The percent reflectance of leaves fits a quartic polynomial model: (1) p=a0+a1λ+a2λ2+a3λ3+a4λ4+cr+lf.

We compared the percent reflectance of each leaf with different exposure times using ANOVA. We carried out subsequent post-hoc tests using the Holm-Sidak multiple comparison test with a significance level of P = 0.05.

For all GC-MS spectra, the internal standard ratio was calculated by taking the total peak area for each compound (counts ×min) and dividing by the peak area of the IS, nonyl acetate (RT = 19.36). We calculated all peak area percentages and IS ratios by taking the mean value of the total number of replicates for each treatment. For each compound, differences between treatments were tested with a Krikal-Wallis test.

Results

Volatile collection

GC-MS analysis yielded several interesting compounds unique to TYLCV-infected tomatoes compared to uninfected control plants (Table 1). Of particular interest is that MeSA, an ester upregulated by many plants in response to biotic stress, increased significantly in late-stage TYLCV infections, faintly in mid-stage and presymptomatic infections, and was absent in controls (Table 1).

In addition, 3,7,7-trimethyl-1,3,5-cycloheptatriene, a homoterpene that was already found to increase in tomatoes following herbivory damage (Ayelo et al., 2021), increased significantly in all TYLCV-infected tomato stages. The production of terpene volatiles, including α-pinene, β-phellandrene, 4-carene, caryophyllene, α-terpinene, and humulene also increased in TYLCV-infected tomatoes compared to controls, but the difference was not statistically significant.

Color spectrometry

We obtained the spectral reflectance data of 64 tomato leaves and a yellow card, including presymptomatic, mid-stage symptomatic, and late-stage symptomatic leaves from tomato plants at one of three stages of TYLCV infection (Fig. 3, Table 2). We measured the spectral reflectance on each of the 32 leaves in the presymptomatic category. The percent reflectance relative to standard on green areas in control leaves was similar to midstage and presymptomatic tomato leaves for long wavelengths, and only with pre-symptomatic for medium wavelengths (Fig. 3A, Table 2). However the main differences in refelectance among mid and late infection stage were found in yellow and dark yellow areas that differed significantly depending on the infection status (up to 33% more reflectant in late stage infection as compared to mid stage) (Table 2). Late-stage leaves had a higher percent reflectance across all wavelength ranges than the others (Fig. 3C, Table 2). The yellow card used as a control had a much higher reflectance than the mid-stage or late-stage symptomatic at all wavelengths (Fig. 3B).

Y-tube choice assays

In control choice assays where all whiteflies and plants were TYLCV-uninfected, B. tabaci showed a preference for volatiles produced by tomatoes over those produced by collard, regardless of host rearing (Fig. 4). There was no difference in host preference overall between the B. tabaci reared on collard compared to B. tabaci reared on tomato (χ2 = 0.065, df = 1, P = 0.799).

Uninfected whiteflies were overall more attracted to TYLCV-infected tomatoes than viruliferous whiteflies independently of the cues (olfactory or visual) used (GLM with binomial distribution, χ2 = 6.516, df = 1, P = 0.011). For experiments using only visual cues, we found that the viral status of the whitefly (TYLCV-infected or uninfected) significantly influenced host choice across all stages of infection (χ2 = 6.03, df = 1, P = 0.025). Viruliferous whiteflies showed no preference for a healthy versus an infected host plant despite a slightly higher choice count towards healthy tomatoes at any stage of the disease (Fig. 5A). Additional analysis revealed that uninfected whiteflies were visually more attracted to TYLCV-infected tomatoes at the mid-stage of infection as compared to uninfected tomato (χ2 = 4.05, df = 1, P = 0.004) (Fig. 5B). TYLCV-infected tomatoes visual cues were more attractive to uninfected whiteflies than to viruliferous whiteflies (Fig. 5C). For experiments using only olfactory cues, whitefly viral status also had a significant influence on all stages of plant infection in the same manner as bioassays with vision only (χ2 = 3.75, df = 1, P = 0.021) where viruliferous whiteflies preferred healthy tomato and uninfected whiteflies preferred TYLCV-infected tomato (Fig. 6C). GLM analysis also found late-stage symptomatic tomato plant volatiles to be more attractive to uninfected whiteflies than healthy tomato volatiles (χ2 = 3.2, df = 1, P = 0.017) (Fig. 6B). Finally, combined visual and olfactory experiments also yielded similar results where uninfected whitefly prefered TYLCV-infected tomato rather than uninfected conterparts (χ2 = 4.15, df = 1, P = 0.027) (Fig. 7B). As seen in olfactory cues only experiments, choice assays that combined both cues saw significant preference of uninfected whiteflies selecting TYLCV-infected tomato at late-stage symptoms (χ2 = 4.05, df = 1, P = 0.004) (Fig. 7C).

Figure 3 Mean spectral reflectance measurements of different tomato leaf areas depending on their infection status.

(A) Mean spectral reflectance measurements of green spots from tomato leaves from one of three different stages of TYLCV (presymptomatic, mid stage symptomatic, and late stage symptomatic). Dash-dot-dash line represents control leaves (n = 6), solid line represents pre-symptomatic (n = 32), dashed line represents mid-stage symptomatic (n = 21), and dotted line represents late-stage symptomatic (n = 12). (B) Mean spectral reflectance measurements of yellow spots from tomato leaves from two different stages of TYLCV (mid stage symptomatic and late stage symptomatic). Control and pre-symptomatic tomatoes do not show yellow spots. The dashed yellow line represents mid-stage symptomatic (n = 20); and the dotted yellow line represents late-stage symptomatic (n = 13). Finally, mean spectral reflectance measurements of yellow sticky card (n = 3). (C) Mean spectral reflectance measurements of dark yellow spots from two different stages of TYLCV (mid-stage symptomatic and late-stage symptomatic). Control and pre-symptomatic tomatoes do not show dark yellow spots. The dashed red line represents mid-stage symptomatic (n = 12), and the dotted red line represents late-stage symptomatic (n = 8).

Table 2 Percent reflectance statistics for leaves of tomato at various stages of TYLCV infection: control, pre-symptomatic, mid-stage, and late-stage infection.

Percent reflectance is the amount of reflection relative to the total light reflected by the reflection standard. The two color ranges compared the percent reflectance measurements every 0.5 nm between 496 and 570.5 nm for green and 571 and 590.5 nm for yellow. We analyzed the values in a linear model that fit the quartic equation. We carried out subsequent post-hoc tests using the Holm-Sidak multiple comparison test with a significance level of P = 0.05. Different letters indicate significantly different percent reflectance values.

Color of leaf area measured according to observer	Plant status
(n = sample size)	Medium and long wavelength
percent reflectance relative to standard
(400–750 nm)	Medium wavelength percent reflectance relative to standard
(496–570.5 nm)	Long wavelength percent reflectance relative to standard
(571–590.5 nm)	
		Mean	±SE	Letters	Mean	±SE	Letters	Mean	±SE	Letters	
Green	Control (n = 6)	25.58	0.12	a	26.8	0.24	b	26.5	1.11	ab	
Green	Pre-symptomatic (n = 32)	26.42	0.05	c	27.1	0.10	b	26.8	1.01	b	
Green	Mid-Stage (n = 21)	25.44	0.06	a	25.4	0.13	a	25.1	1.02	a	
Green	Late-Stage (n = 12)	26.06	0.09	b	25.8	0.17	a	25.8	1.05	ab	
Yellow	Mid-Stage (n = 20)	28.26	0.07	d	29.2	0.13	c	30.8	1.03	c	
Yellow	Late-Stage (n = 13)	34.98	0.08	f	35.5	0.16	d	40.8	1.05	d	
Dark Yellow	Mid-Stage (n = 12)	28.66	0.09	e	27.3	0.17	b	30.0	1.05	c	
Dark Yellow	Late-Stage (n = 8)	36.49	0.10	g	34.7	0.21	d	40.1	1.08	d	

Figure 4 Olfactory Y-tube assays showing the proportion of Bemisia tabaci (uninfected with TYLCV) choosing TYLCV-free tomato volatiles vs. collard volatiles.

Assays were conducted with whiteflies reared on collard (N = 40) and TYLCV-free tomatoes (N = 40).

Figure 5 Visual Y-tube assays showing the percentage of Bemisia tabaci response to visual sources.

(A) TYLCV viruliferous or (B) uninfected choosing TYLCV-infected tomato visual cues over healthy tomato visual cues at three different stages of infection: pre-symptomatic (pre), mid-stage symptoms (mid), and late-stage symptoms (late). Significant differences in host choice between visual targets are indicated (N = 40, α = 0.05). (C) Comparison of TYLCV viruliferous vs. uninfected whitefly preference for TYLCV-infected tomato based on visual cues. Decision threshold over 50% indicates preference for TYLCV-infected tomato as indicated by the black midway line.

Figure 6 Olfactory Y-tube assays showing the percentage of Bemisia tabaci response to odor sources.

(A) TYLCV viruliferous or (B) uninfected choosing TYLCV-infected tomato olfactory cues over healthy tomato olfactory cues at three different stages of infection: pre-symptomatic (pre), mid-stage symptoms (mid), and late-stage symptoms (late). Significant differences in host choice based on olfactory cues are indicated (N = 40, α = 0.05). (C) Comparison of TYLCV viruliferous vs. uninfected whitefly preference for TYLCV-infected tomato based on olfactory cues. Decision threshold over 50% indicates preference for TYLCV-infected tomato as indicated by the black midway line.

Figure 7 Vision + olfaction Y-tube assays showing the percentage of Bemisia tabaci response to odor and visual sources.

(A) TYLCV viruliferous or (B) uninfected choosing TYLCV-infected tomato visual + volatile cues over healthy tomato visual + volatile cues at three different stages of infection: pre-symptomatic (pre), mid-stage symptoms (mid), and late-stage symptoms (late). Significant differences in host choice between visual targets are indicated (N = 40, α = 0.05). (C) Comparison of TYLCV viruliferous vs. uninfected whitefly preference for TYLCV-infected tomato based on visual and olfactory cues. Decision threshold over 50% indicates preference for TYLCV-infected tomato as indicated by the black midway line.

We exposed uninfected whiteflies to different concentrations of MeSa. The concentrations of methyl salicylate that were not more attractive than the control were 0.1 µg/ µL (χ2 = 1.23, df = 1, P = 0.26, Fig. 8B) and 1 µg/ µL (χ2 = 0.03, df = 1, P = 0.87, Fig. 8B), while 0.01 µg/ µL was more attractive compared to the control (χ2 = 7.81, df = 1, P < 0.01, Fig. 8B). Viruliferous whiteflies did not show preference to different concentrations of methyl salicylate compared to the control for 1 µg/ µL (χ2 = 3.46, df = 1, P = 0.06, Fig. 8A), 0.1 µg/ µL (χ2 = 3.27, df = 1, P = 0.07, Fig. 8A), or 0.01 µg/ µL (χ2 = 1.32, df = 1, P = 0.25, Fig. 8A).

Figure 8 Olfactory Y-tube assays showing the percentage of Bemisia tabaci response to different Methyl salicylate (MESA) dosages.

(A) TYLCV viruliferous or B) uninfected choosing different concentrations of MESA (1.0, 0.1, 0.01 ng/µl) vs control (mineral oil). Significant differences in host choice between MESA concentrations are indicated (N = 40, α = 0.05).

Two choice visual assay

The intensity of the green visual target was 94.39 ± 0.22 µwatts/cm2/sond relative was adjusted to be nearly equal to the yellow visual target, which was 92.53 ± 0.07 µwatts/cm2/sond using neutral density filters. The total response of whiteflies to either filter was not significant (χ2 = 16.91, df = 11, P = 0.11). Whitefly response to the visual target was not affected by filter placement randomly on the left or right side (χ2 = 0.22, df = 1, P = 0.64) or sex of the whitefly (χ2 = 2.39, df = 2, P = 0.30). However, the whiteflies’ response to visual targets significantly differed based on whether researchers reared them on a TYLCV positive or negative host plant (χ2 = 6.72, df = 1, P < 0.01). Whiteflies reared on TYLCV-negative collards preferred green visual targets over yellow ones (χ2 = 204, df = 144, P < 0.001, Fig. 9A). In contrast, whiteflies reared on TYLCV-positive tomatoes show no preference between yellow or green visual targets (χ2 = 288, df = 256, P = 0.08, Fig. 9B).

Figure 9 Two-choice visual assays showing the percentage of Bemisia tabaci response to yellow and green visual targets.

(A) TYLCV uninfected or (B) viruliferous choosing green or yellow targets. Significant differences in host choice between MESA concentrations are indicated by different letters (N = 25, α = 0.05).

Discussion

As with all insect pests, the management of agricultural pests benefits from understanding the drivers of pest behavior. In the case of whiteflies and TYLCV, there is a complex world of visual and chemical cues that influence B. tabaci, which the virus can utilize either directly or indirectly (Maluta, Fereres & Lopes, 2019). A previous study found that B. tabaci that acquired TYLCV had a higher metabolism, corresponding to decreased longevity (Pusag et al., 2012).

Viruliferous whiteflies demonstrate direct manipulation by feeding more frequently on phloem sieve tube elements and exhibiting an extended salivation phase before phloem-feeding, which enhances the virus’s inoculation (Moreno-Delafuente et al., 2013). An example of indirect manipulation occurs when TYLCV enhances a host plant’s attraction to non-viruliferous whiteflies, leading to increased settlement on the plant, faster probing, and a greater number of feeding bouts (Liu et al., 2013). TYLCV may do this by phenotypically altering the host plant through visual changes (i.e., yellowing/mottling), changes in emitted volatiles, or a combination of both. Despite a general understanding of this disease cycle, the exact interplay between TYLCV manipulation and B. tabaci host selection has yet to be defined.

We found that the most significant influence on B. tabaci host choice was the TYLCV infection status of the whitefly with uninfected B. tabaci selecting the olfactory and visual cues of TYLCV positive host plants in y-tube assays. These findings contrast with those of Fereres et al. (2016), who reported that B. tabaci preferred uninfected tomato plants when infected with Tomato Severe Rugose Virus (ToSRV), a circulative-transmitted begomovirus. This discrepancy could be attributed to the differing volatile emission profiles of tomatoes infected with ToSRV compared to those infected with TYLCV in our study. Another key difference between Fereres et al. (2016) and our research is the duration of whitefly exposure to the virus. In the study by Fereres et al. (2016) whiteflies were exposed to ToSRV-infected plants for 72 h, whereas our study involved whiteflies that were reared on TYLCV-positive plants and maintained on TYLCV-positive plants as adults. Since TYLCV is a persistent, circulative virus, the findings of this study are consistent with the mechanism of viral propagation. It is advantageous for a virus such as TYLCV to enhance the attractiveness of a plant host to a healthy insect vector and induce behavioral modifications in the insect that make infected plants less attractive once viral acquisition by the vector has occurred. The viral-induced changes to the plant increase the likelihood of the insect vector dispersing to another host in search of more favorable nutrition, infecting new plants and continuing the disease cycle.

The stage of TYLCV infection in the host tomato plant also played a role in attraction and host selection for B. tabaci and is further illuminated by color spectrometry data. In presymptomatic stages of TYLCV infection (2 weeks post-inoculation), there were no visible signs of disease and no significant difference between whiteflies choosing this stage versus healthy tomatoes. Once plants had reached mid-stage TYLCV infections (1-month post inoculation), tomatoes were significantly more attractive to healthy whiteflies via visual symptoms only. For mid-stage and late-stage TYLCV infections, visible yellowing and curling of leaves had begun to occur, exhibiting higher reflectance in both the green and yellow wavelengths. This change in the visible spectrum was likely more attractive to B. tabaci since yellow wavelengths are desirable (Gu et al., 2008). The increased percent reflection of tomato leaves as the disease progressed in our study may further attract whiteflies to diseased plants compared to lesser reflective leaves of undiseased plants. Lu et al. (2018) found that TYLCV-infected tomatoes had increased reflection in the visible spectrum at 550 and 600 nm. Variations in water, chlorophyll, or other molecule content in the leaves may cause shifts in reflectance (Hunt, Rock & Nobel, 1987; Montasser et al., 2012; Calderón et al., 2013; Khalil et al., 2014). These findings concur with previous studies that have found that the altered phenotype induced by geminiviruses in a host plant can increase the visual attraction of its insect vector (Lu et al., 2019; Wang et al., 2019).

Surprisingly, our visual two-choice assay indicated that the virus alters the color preference of whiteflies so that viruliferous whiteflies have no preference between green and yellow visual targets, whereas uninfected whiteflies preferred the green target. In a previous study it was found that in a non choice test, TYLCV-infected whiteflies increased their response to green LEDs compared to uninfected whiteflies (Jahan et al., 2014). However, in the two choice experiment conducted we found that viruliferous whiteflies showed no preference between yellow vs green targets as compared to uninfected whiteflies that chose green targets. The result of our choice test may indicate an increase in attraction for yellow color (rather than a decrease in attraction to green color). The shift in preference was only evident for visual targets in the visible spectrum (400–700 nm) with a focus on medium (green) and longer wavelengths (yellow), as those constitute the range of the electromagnetic spectrum characterizing tomato leaves. Future studies should evaluate ultraviolet light’s effect on visual targets. For instance, Johnston & Martini (2020) found that uninfected whiteflies preferred yellow targets to green targets represented as colored cards and UV were not excluded. It is possible that the exclusion of UV may explained the discrepancy between the two study. Indeed, previous studies have identified ultraviolet light as a potential flight-stimulating wavelength for greenhouse whiteflies (Trialeurodes vaporariorum) (Coombe, 1981). Several studies indicated that the combination of green or yellow and ultraviolet wavelengths increased the attraction of whiteflies to visual targets (Vaishampayan et al., 1975; Stukenberg, Gebauer & Poehling, 2015). Conversely, the combination of blue and ultraviolet LEDs decreased the number of greenhouse whiteflies on host plants (Athanasiadou & Meyhöfer, 2023).

The modulation of volatile profiles in infected plants also changes the host selection behavior of insect vectors (Chen et al., 2017; Mwando et al., 2018). Despite the importance of visual cues in host selection for B. tabaci, olfactory cues also provided an important added stimulus, as evidenced by uninfected whiteflies choosing TYLCV-infected plants at late stages of infection. Analysis of volatile profiles at different stages of infection in tomatoes using GC-MS offers some insights into specific emitted compounds that elicit an increased attractive response in the whitefly. Many of the selected peaks, such as α-pinene, cymene, carene, and γ-terpinene are terpenoids produced by tomato plants to discourage herbivory and act as repellents (Chen et al., 2017; Zhang et al., 2022). TYLCV-infected plants initially had higher concentrations of these compounds in presymptomatic plants, which gradually became lower in the mid and late stages of infection. Lower terpenoid concentrations in TYLCV-infected plants have been shown to encourage B. tabaci in selecting and feeding on hosts (Luan et al., 2013). Therefore, it is likely that tomato plants produce higher concentrations of terpenes after initial inoculation with TYLCV as a defense mechanism. The plant defense mechanism may be a factor that contributes to reduce attraction of TYLCV-tomato that are in pre- and mid-symptomatic stages (Figs. 6 and 7). Another example of altering whitefly host selection through the tomato’s volatile profile is the production of MeSA, an ester that many plants produce in response to biotic stress. Methyl salicylate, a compound absent in healthy control plants, was detected in higher amounts in late-stage infections as compared to earlier infection stages (Table 1). Closely related insects such as the Asian citrus psyllid, Diaphorina citri, have been found to show increased attraction to citrus trees producing higher quantities of methyl salicylate after conspecific damage or after infections of the bacteria causing citrus greening (Mann et al., 2012; Martini et al., 2016). In the same way, TYLCV-induced production of methyl salicylate in tomatoes may serve as a signal to foraging whiteflies, enhancing the tomato plant’s attractiveness as a host. Our lab-based data indicated that only uninfected whiteflies were more attracted to higher concentrations of MeSA, whereas viruliferous whiteflies showed no preference. The modulation of MeSA supports the concept that the virus indirectly attracts nonvirus-carrying whiteflies by the olfactory cues it produces. Viruliferous whiteflies may exhibit reduced sensitivity to MeSA due to the BtPMaT1 gene, which allows them to detoxify phenolic glycosides and neutralize defense signals in host plants (Xia et al., 2021). Whitefly infestations can alter plant volatile emissions and the production of SA and JA, further interfering with plant defenses (Zhang et al., 2009; Zhang et al., 2019). Additionally, TYLCV can manipulate B. tabaci to be attracted to odors and visual cues normally less attractive to non-viruliferous whiteflies, which, combined with B. tabaci’s ability to neutralize and manipulate plant defenses, may contribute to reduced sensitivity to MeSA.

These findings suggest that TYLCV manipulates B. tabaci both directly and indirectly. Direct manipulation occurs when the virus decreases the attraction of infected whiteflies toward infected plants that the insects would otherwise find highly attractive. As a result, infected B. tabaci disperse, increasing the likelihood of settling on a healthy, uninfected plant. Indirect manipulation occurs when the virus increases the attractiveness of visual and olfactory cues emitted by infected plants towards uninfected whiteflies, allowing the viral acquisition to appear in a fresh generation of insect vectors. In light of current knowledge about the cycle of TYLCV in both tomato and whitefly hosts, many questions remain unanswered. One area deserving further scrutiny is how the virus alters the plant host’s nutritional quality at different infection stages. Researchers have already found that TYLCV enhances the nutritional quality of leaf tissue and phloem sap for B. tabaci while simultaneously reducing the production of defensive, anti-herbivory compounds (Su et al., 2015); however, the degree and length of time these changes occur in the disease cycle has yet to be quantified. These findings should contribute to a better understanding of the interaction between whiteflies, plants, and pathogens, allowing for more precise insect management strategies and prevention of further associated disease outbreaks.

Supplemental Information

Supplemental Information 1 Two Choice Visual Assay

Supplemental Information 2 Reflectance mid symptomatic leaves yellow areas

Supplemental Information 3 Reflectance mid symptomatic leaves dark yellow areas

Supplemental Information 4 Reflectance late symptomatic leaves dark yellow areas

Supplemental Information 5 Reflectance late symptomatic leaves green areas

Supplemental Information 6 Reflectance pre symptomatic leaves green areas

Supplemental Information 7 Visual, odor and visual + odor Y tube tests

Supplemental Information 8 Reflectance late symptomatic leaves yellow areas

Supplemental Information 9 Reflectance uninfected (control) leaves green areas

Supplemental Information 10 Reflectance mid symptomatic leaves green areas

Supplemental Information 11 Y-tube assay with MeSA

We want to thank Fanny Iriarte and all the support from the plant pathogen diagnostics lab at the North Florida Research and Education Center. The contents do not necessarily reflect the views or policies of the U.S. Department of Agriculture and of the National Science Foundation nor does mention of trade names, commercial productions, services or organization imply endorsement by the U.S. government.

Additional Information and Declarations

Competing Interests

Author Contributions

Data Availability

The authors declare there are no competing interests.

Thomson M. Paris conceived and designed the experiments, performed the experiments, analyzed the data, prepared figures and/or tables, authored or reviewed drafts of the article, and approved the final draft.

Nicholas Johnston conceived and designed the experiments, performed the experiments, analyzed the data, prepared figures and/or tables, authored or reviewed drafts of the article, and approved the final draft.

Iris Strzyzewski conceived and designed the experiments, performed the experiments, analyzed the data, authored or reviewed drafts of the article, and approved the final draft.

Jessica L. Griesheimer performed the experiments, authored or reviewed drafts of the article, and approved the final draft.

Benjamin Reimer performed the experiments, authored or reviewed drafts of the article, and approved the final draft.

Kathi Malfa performed the experiments, authored or reviewed drafts of the article, and approved the final draft.

Sandra A. Allan conceived and designed the experiments, authored or reviewed drafts of the article, provided materials and funding, and approved the final draft.

Xavier Martini conceived and designed the experiments, analyzed the data, prepared figures and/or tables, authored or reviewed drafts of the article, provided funding, and approved the final draft.

The following information was supplied regarding data availability:

The raw data are available in the Supplemental Files.

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
