# Peer review of "Tomato yellow leaf curl virus manipulates Bemisia tabaci, MEAM1 both directly and indirectly through changes in visual and volatile cues"

_PeerJ, doi:10.7717/peerj.17665_

## Round 0.1 · original submission · Minor Revisions

Kindly address the issues raised by reviewers.

Reviewer 1 ·

Basic reporting

no comments

Experimental design

no comments

Validity of the findings

no comments

Additional comments

The manuscript entitled ‘Tomato Yellow Leaf Curl Virus manipulates Bemisia tabaci, MEAM1 both directly and indirectly through changes in visual and volatile cues’ has been reviewed. The topic is relevant and important as B. tabaci is one of the most devastative pests and an important vector for most serious damaging group of plant viruses (Geminiviruses) for vegetable crop worldwide.
In this study, authors evaluate how the presence of TYLCV, either in tomato or in whitefly, impacts the host choice and behaviours of B. tabaci. The impact of changes in tomato volatiles and whitefly’s host choices over time studied. Authors conclude that TYLCV manipulates B. tabaci both directly and indirectly and decreases the attraction of infected whiteflies toward infected plants that the insects would otherwise find highly attractive. The results of this study should helpful to a better understanding of the interaction between vector (B.tabaci), host (tomato), and pathogen (TYLCV) and the disease epidemiology which would contribute for developing better insect-vector-virus management strategies.
Manuscript is well organized and well written with sufficient details. The methods are clear and describe in details. Results are well supported with appropriate figures and tables. Results are discussed well however it would be better if little more explanation/discussion provided with following points:
(i) According to the study conducted by Fereres et al. (2016) on interaction of B. tabaci and a begomovirus (Tomato severe rugose virus) preferred virus-infected tomato hosts regardless of whether the whiteflies themselves were infected with the virus which is contrast to your finding. In both cases the B. tabaci genetic grouping is the same i.e. MEAMI and the both cases virus belong to Geminivius group with persistent and calculative type of transmission. What could be the scientific explanation behind the contrasting results? It would be good to extend the discussion further on this point.

(ii) Why do you think that MeSA concentration has no impact on manipulating choices of viruliferous whiteflies? Or other factor involved in combination?

(iii) Do you think that other volatiles have any role to play in modifying whitefly behaviours/ host choices in combination with MeSA? It is not clear as the role of other volatile as data is not given or discussed.

Check all the references carefully specifically the last one, Zhang et al (2021).

Reviewer 2 ·

Basic reporting

The article is written in clear and professional English. Some references to previous work could be added (see comments). The structure of the article is completely standard. The authors pose questions and hypotheses clearly and set up adequate experiments to answer them.

Experimental design

In this article, Paris et al. set up experiments to identify cues in TYLCV-infected plants (at different stages of infection) that induce attraction of the whitefly vector, Bemisia tabaci. They also studied the impact of the viruliferous status of vector whiteflies on their plant selection behaviour. They set up experiments to test the choice of whiteflies and analyzed phenotypic changes in tomatoes, namely color by spectrophotometry and volatile emission by GC-MS. The experiments are well-designed, and the interpretations and statistical analyses of the results are perfectly intelligible. This article presents a high-quality work that confirms observations presented in previous works (by the same team or other international teams) and completes our understanding of the mechanisms involved in the attraction of vectors by infected plants.

Validity of the findings

The data was provided by the authors. To the best of my knowledge, it seems to me that the data are robust, with a sufficient number of replications (at least standard compared to similar papers) and under well-controlled conditions.The authors make good reference to the literature in their discussion. They reposition their results well in relation to previous research and discuss descrepencies.The conclusions are moderate and not overstated compared to their observations.

Additional comments

I have only minor suggestions/comments to clarify a few points.
On a few occasions, the authors use the term "infection status of B. tabaci" (e.g. L. 27-28). As the virus is not (with certainty) multiplicative/propagative for whiteflies (some recent articles suggest it could be), it would be more prudent to use the terms viruliferous/non-viruliferous throughout the manuscript.
It would be useful to mention the following references in the introduction: Ban et al. (2021) (10.1093/jee/toaa326) and Ontiveros et al. (2022) (10.1094/PHYTO-08-21-0341-R), both of which work with the same pathosystem (and other viruses), and to discuss the results in light of this work.

L. 108. Remove the comma after “field”.
L. 120-121. The formulation is not clear to me.
L. 129. Can you describe a bit more precisely what are the “visual targets” mentioned? In Johnston & Martini (2020), two types are used: plant leaves and “artificial” ones. Note that the ref is wrongly cited : (Johnston et al. 2020) should be (Johnston & Martini, 2020).
As I mentioned the references, please pay attention to the list of references, three are repeated: Fang et al., Liu et al., Moreno et al. There might be other mistakes that need to be corrected before publication.
L. 162. Is the Y-tube positioned vertically? In the set-up described in Johnston & Martini (2020), the y-tube is horizontal, right?
L. 196. I'm not competent to judge the experimentation on visual preferences and color spectrometry analysis. However, the description of the protocol seems exhaustive and comprehensive. Just a small question though, isn't the regression coefficient (x) supposed to be present in the quartic polynomial model? Can you justify the use of "Holm-Sidak" multiple comparisons (which I've never heard of)?
L. 324. I'm not sure what this 1st sentence refers to. Could you clarify? Is it a combination of all the data?
The title of Figure 5 seems wrong: “response to odor sources” instead of “response to visual sources”.
In the results section, for figures 5 and 8, the authors present the results of panels C then B then A. It would be great to reorganize the panels or the results section so that the order is restored.
Figure 6C. The legend has shifted titles and the line (decision threshold) is not at 50%.
Figure 7B. The statistical difference indicated in the text (L. 341) is not visualized in the figure.
L. 362. “betwwen” instead of “between”.
L. 386-389. The authors could cite the recent work of K. Mauck on this point, and possibly mention the associated hypothesis of "host and vector manipulation by plant viruses".
L. 417. “coulor” instead of “colour”

---

## Round 0.2 · accepted · Accept

I have reviewed the revised version of the manuscript and the authors' responses to the reviewers' comments. The revised manuscript can be accepted. Both reviewers suggested minor revisions, and I do not feel a second round of review is necessary.